# Serum Metabolomics Reveals a Potential Benefit of Methionine in Type 1 Diabetes Patients with Poor Glycemic Control and High Glycemic Variability

**DOI:** 10.3390/nu15030518

**Published:** 2023-01-19

**Authors:** Liyin Zhang, Keyu Guo, Qi Tian, Jianan Ye, Zhiyi Ding, Qin Zhou, Xia Li, Zhiguang Zhou, Lin Yang

**Affiliations:** National Clinical Research Center for Metabolic Diseases, Key Laboratory of Diabetes Immunology, Ministry of Education, Department of Metabolism and Endocrinology, The Second Xiangya Hospital of Central South University, Changsha 410011, China

**Keywords:** type 1 diabetes, glycemic variability, continuous glucose monitoring, metabolomics

## Abstract

Glycemic variability (GV) in some patients with type 1 diabetes (T1D) remains heterogeneous despite comparable clinical indicators, and whether other factors are involved is yet unknown. Metabolites in the serum indicate a broad effect of GV on cellular metabolism and therefore are more likely to indicate metabolic dysregulation associated with T1D. To compare the metabolomic profiles between high GV (GV-H, coefficient of variation (CV) of glucose ≥ 36%) and low GV (GV-L, CV < 36%) groups and to identify potential GV biomarkers, metabolomics profiling was carried out on serum samples from 17 patients with high GV, 16 matched (for age, sex, body mass index (BMI), diabetes duration, insulin dose, glycated hemoglobin (HbA1c), fasting, and 2 h postprandial C-peptide) patients with low GV (exploratory set), and another 21 (GV-H/GV-L: 11/10) matched patients (validation set). Subsequently, 25 metabolites were significantly enriched in seven Kyoto Encyclopedia of Genes and Genomes (KEGG) pathways between the GV-H and GV-L groups in the exploratory set. Only the differences in spermidine, L-methionine, and trehalose remained significant after validation. The area under the curve of these three metabolites combined in distinguishing GV-H from GV-L was 0.952 and 0.918 in the exploratory and validation sets, respectively. L-methionine was significantly inversely related to HbA1c and glucose CV, while spermidine was significantly positively associated with glucose CV. Differences in trehalose were not as reliable as those in spermidine and L-methionine because of the relatively low amounts of trehalose and the inconsistent fold change sizes in the exploratory and validation sets. Our findings suggest that metabolomic disturbances may impact the GV of T1D. Additional in vitro and in vivo mechanistic studies are required to elucidate the relationship between spermidine and L-methionine levels and GV in T1D patients with different geographical and nutritional backgrounds.

## 1. Introduction

Type 1 diabetes (T1D) is characterized by intrinsic immune destruction of β-cell, and insulin-dependent treatment to achieve euglycemia results in substantial glycemic variability (GV) [1]. In patients with T1D, an accurate and complete GV assessment is critical for evaluating recent glucose control and the long-term risk of chronic complications. Although glycated hemoglobin (HbA1c) has been recognized as the “gold standard” for assessing glycemic control, it does not reflect intra- and interday GV that gives rise to acute events or postprandial hyperglycemia related to both microvascular and macrovascular complications. Improving HbA1c significantly reduces the risk of target-organ complications [2,3]. However, macrovascular complications continue, albeit at lower rates [4], which might be related to GV. Given the close correlation between GV and diabetic complications and hypoglycemia, the international consensus on the use of continuous glucose monitoring (CGM)-issued guidelines for the management of GV recommended that the coefficient of variation (CV) of glucose should be <36% [5].

Several clinical factors, such as diabetes duration, insulin dose, HbA1c levels, and C-peptide levels, may affect GV. Among them, pancreatic islet function markers are the most relevant [6,7]. Surprisingly, a considerable variation was detected in GV, and whether other factors are involved is yet unknown, although the levels of the aforementioned GV-affecting clinical indicators of some T1D patients were comparable. This phenomenon may be explained by metabolomics, a rapidly evolving technology that quantifies all endogenous metabolites in cells or tissues after genetic or environmental intervention [8]. The metabolites in serum indicate a broad effect of GV on cellular metabolism and therefore are likely to indicate metabolic dysregulation associated with T1D. Therefore, the serum metabolome contains critical biomarkers of GV, which reveals the potential pathways involved in the pathogenesis of complications in T1D and yields novel therapeutic targets.

The metabolic response to glycemic control in T1D patients is complicated. Despite good glycemic control (HbA1c < 6.5%), the metabolic profile of T1D patients was significantly altered compared to healthy controls without diabetes [9]. Furthermore, branched-chain amino acids (BCAAs) and alanine levels were found to be associated with blood glucose levels in T1D patients [10]. Although good glycemic control normalized some metabolic pathways, a large number of other metabolic pathways remained abnormal, indicating that T1D patients exhibited a variety of metabolic disorders.

Therefore, we aimed to use untargeted metabolomics analysis to identify metabolic disorders associated with GV in T1D patients and identify potential GV biomarkers after matching for common clinical factors affecting GV. To the best of our knowledge, the metabolomic profiling of GV in T1D patients has not yet been investigated.

## 2. Materials and Methods

### 2.1. Study Participants

A total of 54 T1D patients from the Second Xiangya Hospital of Central South University from November 2019 to February 2022 were included. The 1999 World Health Organization criteria for diabetes diagnosis were used. The inclusion criteria were as follows: (1) acute diabetes onset; (2) insulin-dependent from the time of diabetes onset. The exclusion criteria were as follows: (1) other diabetes types; (2) acute diabetic complications such as diabetic ketoacidosis; (3) acute infectious diseases, recent surgery, trauma, or more stressful situations; (4) use of glucocorticoids or immunomodulators; (5) pregnancy or preparation for pregnancy; (6) a comorbid autoimmune disease.

Finally, 33 patients comprised the exploratory set and were divided into two groups: those with high GV (GV-H, *n* = 17) and those with low GV (GV-L, *n* = 16). GV-H was defined as a glucose CV ≥ 36%, which reflects the magnitude of the GV as reported previously [5,11]. Thus, GV-L was defined as having a glucose CV < 36%. The two groups were further matched for age, gender, body mass index (BMI), duration of diabetes, daily insulin dosage, fasting blood glucose (FBG), 2 h postprandial blood glucose (2hBG), HbA1c, fasting C-peptide (FCP), and 2 h postprandial C-peptide (2hCP). Subsequently, another 21 subjects (GV-H group, *n* = 11; GV-L group, *n* = 10) were recruited to form the validation set using the same inclusion, exclusion, and matching criteria, as described above. Together, 54 subjects were included in the present study.

The study protocol was approved by the Ethics Review Committee of the Second Xiangya Hospital of Central South University (No. 2019-186; Date: 12 November 2019) and was conducted in compliance with the Declaration of Helsinki. Each subject provided signed informed consent.

### 2.2. Data Collection and Biochemical Measurements

At the time of enrolment, demographic and clinical data such as age, gender, height, weight, BMI, waist/hip circumference, blood pressure, time of diabetes onset, diabetic ketoacidosis at onset, and current insulin schema were collected. Height and weight were measured using the electronic stadiometer and weighing scale. BMI was calculated as weight in kg/(height in m)^2^. Patients were asked to stand with their arms crossed on the contralateral shoulders for waist circumference measurement. The measuring tape should be snug around the lateral aspect of each ilium at the mid-axillary line. The patients were asked not to smoke or drink strong tea or coffee within 30 min before blood pressure measurement, and to empty their bladder. Blood pressure was measured using cuff pressurization after sitting in a chair with a backrest in a quiet environment for at least 5 min. Fasting venous blood samples were drawn from all study participants in the morning after they had fasted for 10 h for the following tests: HbA1c, FBG, FCP, and lipid profiles. Blood samples for 2hBG and 2hCP testing were taken 2 h following a mixed-meal tolerance test (MMTT; 44.4% carbohydrates, 47.7% fat, and 7.9% protein).

Automated high-performance liquid chromatography was used to measure HbA1c levels (VARIANT II Haemoglobin Testing System; Bio-Rad Laboratories), which is the gold standard method for monitoring glucose control in diabetes patients as recommended by the American Diabetes Association (ADA) [12]. The serum CP levels were determined using a chemiluminescent method and an Adivia Centaur XP immunoassay system (Siemens, Germany). The inter-assay and intra-assay variation coefficients of the CP test were 3.7–4.1% and 1.0–3.3%, respectively.

### 2.3. Continuous Glucose Monitoring

The blinded CGM system was used to generate dynamic glucose profiles (iPro2 with Enlite sensor, Medtronic MiniMed, Northridge, CA, USA). The CGM system’s glucose sensor (MMT-7008A) was implanted on the lateral upper arm and removed after one week, generating a maximum daily record of 288 continuous sensor glucose measurements. Self-monitoring of blood glucose (SMBG) was required for the participants in order to calibrate CGMs at least four times/day. Then, the CGM data were exported and subjected to quality assessment. When at least 70% of the CGM data (5 valid days, equivalent to 1440 glucose readings) was available, the CGM parameters were calculated using M-Smart software (CareLink iPro) provided by Medtronic.

### 2.4. Sample Preparation and Metabolomic Analysis

In positive and negative ESI modes, serum samples were analyzed using Liquid Chromatography Mass Spectrometer (LC-MS). The extraction of metabolites, instrument settings, peak intensities of metabolites, and differential regulation of metabolites between groups were determined as previously described [13,14]. First, the sample was thawed at 4 °C, vortexed for 1 min, and mixed evenly. Second, an appropriate amount of the sample was transferred into a 2 mL centrifuge tube. Third, 400 µL of methanol (stored at −20 °C) was added and vortexed for 1 min. Then, the supernatant was collected by centrifugation of the sample at 12,000 rpm, 4 °C for 10 min; it was further concentrated and dried. Finally, 150 µL of 2-chloro-l-phenylalanine (4 ppm) solution prepared with 80% methanol water (stored at 4 °C) was added to resolubilize the sample, and the supernatant was filtered through a 0.22 μm membrane and transferred into the detection bottle for LC-MS detection.

LC analysis was performed on a Vanquish UHPLC System (Thermo Fisher Scientific, Waltham, MA, USA). The separation was conducted under the following gradient: 0–1 min, 2% C; 1–9 min, 2–50% C; 9–12 min, 50–98% C; 12–13.5 min, 98% C; 13.5–14 min, 98–2% C; 14–20 min, 2% C. For LC-ESI (-)-MS analysis, the analytes were carried out with (A) acetonitrile and (B) ammonium formate (5 mM). Chromatography was performed on a Waters ACQUITY UPLC^®^ HSS T3 (150 × 2.1 mm, 1.8 µm) column maintained at 40 °C. The injection volume and flow rate were set at 0.25 mL/min and 2 L, respectively. The mobile phases for LC-ESI (+)-MS analysis were (C) 0.1% formic acid in acetonitrile (*v*/*v*) and (D) 0.1% formic acid in water (v/v). The gradient was as follows: 0–1 min, 2% C; 1–9 min, 2–50% C; 9–12 min, 50–98% C; 12–13.5 min, 98% C; 13.5–14 min, 98–2% C; 14–20 min, 2% C. The analytes for LC-ESI (-)-MS analysis were (A) acetonitrile and (B) ammonium formate (5 mM). The gradient used for the separation was as follows: 2% A for 0–1 min; 2–50 min; 50–98 min; 12–13.5 min; 98% A; 13.5–14 min; 98–2% A; and 2% A for 1–17 min. Metabolites were detected using mass spectrometry on an Orbitrap Exploris 120 (Thermo Fisher Scientific) with an ESI ion source. The simultaneous acquisition of MS1 and MS/MS (full MS-ddMS2 mode, data-dependent MS/MS) was employed. The parameters were as follows: sheath gas pressure, 30 arb; aux gas flow, 10 arb; spray voltage, 3.50 kV and −2.50 kV for ESI (+) and ESI (−), respectively; capillary temperature, 325 °C; MS1 range, *m*/*z* 100–1000; MS1 resolving power, 60,000 FWHM; number of data-dependent scans per cycle, 4; MS/MS resolving power, 15,000 FWHM; normalized collision energy, 30%; dynamic exclusion time, automatic.

To correct for any systematic bias, the robust locally estimated scatterplot smoothing (LOESS) signal correction (QC-RLSC) was used for data normalization [15]. To guarantee accurate metabolite identification, only ion peaks with relative standard deviations (RSDs) 30% in quality control (QC) were maintained following normalization. The raw UPLC Q-TOF MS data were first converted to mzXML format using MSConvert in the ProteoWizard software program (v3.0.8789) before being processed with XCMS for feature detection, retention time correction and alignment, peak discrimination, filtering, alignment, matching, and identification. Then, a peak table with mz value, retention time (rt), and the corresponding peak intensity was generated. The metabolites were identified by accuracy mass (<30 ppm) and MS/MS data that were compared to multiple databases. Metabolite identification was first confirmed based on the exact molecular weight and subsequently on the Human Metabolome Database (HMDB) according to the MS/MS fragmentation patterns (http://www.hmdb.ca (accessed on 15 March 2022)), massbank (http://www.massbank.jp/ (accessed on 15 March 2022)), LipidMaps (http://www.lipidmaps.org (accessed on 15 March 2022)), mzcloud (https://www.mzcloud.org (accessed on 15 March 2022)), and Kyoto Encyclopedia of Genes and Genomes (KEGG) databases (https://www.genome.jp/kegg/ (accessed on 15 March 2022)). Finally, the metabolites were validated by comparing retention time and fragmentation pattern to authentic standards. Specifically, metabolites were first identified by the exact molecular weight (primary annotation) and then obtained by matching MS/MS fragments with the Human Metabolome Database (HMDB, http://www.hmdb.ca (accessed on 15 March 2022)), MassBank (http://www.massbank.jp/ (accessed on 15 March 2022)), LipidMaps (http://www.lipidmaps.org (accessed on 15 March 2022)), mzCloud (https://www.mzcloud.org (accessed on 15 March 2022)), and a self-built standard database BioDeepDB (PANOMIX, Suzhou, China) (secondary annotation). Differential metabolites were sought from the sample primary substance list and screened at the preset *p*-value and variable importance on projection (VIP) thresholds in the statistical test. Subsequently, a total of 5597 metabolites were identified in the primary and secondary annotations (pos: 3012, neg: 2585), and 569 metabolites were identified in the secondary annotation. All subsequent analyses were based on the results of the secondary annotation. 

### 2.5. Statistical Analysis

Independent sample t-test, Mann–Whitney *U* test, and χ^2^ test were used to compare the basic characteristics between GV-H and GV-L groups for normally distributed, skewed-distribution, and categorical variables, respectively. 

The ropls R package from the Bioconductor repository was utilized for all multivariate data analyses and modeling [16]. Data were mean centered using scaling. Models were constructed using principal component analysis (PCA), partial least-square discriminant analysis (PLS-DA), and orthogonal partial least-square discriminant analysis (OPLS-DA). The VIP derived by OPLS-DA was used to screen out differential metabolites between the GV-H and GV-L groups. In addition, the metabolites identified by the OPLS-DA model were confirmed at a univariate level using the Wilcoxon–Mann–Whitney test (*p* < 0.05). The fold change (FC) was determined according to the relative quantification reflected by the peak intensity of the metabolites between the two groups (GV-H vs. GV-L). Finally, those with *p* < 0.05 and VIP > 1 were considered to be statistically significant metabolites. MetaboAnalyst, which combines the outcomes of potent pathway enrichment analysis and pathway topology analysis, was used to conduct pathway analysis on various metabolites. To adjust for confounding factors, a multiple regression analysis was conducted.

Spearman’s correlation analysis was performed to evaluate the correlation between final selected differential metabolites and glycemic parameters. Multiple linear regression analysis was utilized to explore the relationship of glucose CV with age, BMI, duration of diabetes, FBG, HbA1c, daily insulin dosage, FCP, 2hCP, and final selected potential biomarkers by employing a stepwise procedure, which has been shown to be efficient in picking independent variables that are truly useful for predicting CV since it automatically eliminates the collinearity overstatement. The glucose CV was regarded as the dependent variable, while the other factors were regarded as independent variables. Receiver operating characteristics (ROC) curve analysis was used to determine the predictive ability of differential metabolites for GV and their performance as biomarkers. A two-tailed test was performed, and *p* < 0.05 was considered statistically significant. All statistical analyses were carried out using SPSS 26.0 software (IBM Corp., Armonk, NY, USA).

## 3. Results

### 3.1. Comparison of Basic Characteristics and CGM Parameters of Study Subjects

A total of 33 patients with T1D were divided into GV-H (glucose CV ≥ 36%, *n* = 17) and GV-L (glucose CV < 36%, *n* = 16) groups based on the CV of glucose obtained from CGM (Table 1). The mean glucose CV in the GV-H group was 46.3 ± 5.7% and 26.8 ± 3.7% in the GV-L group, respectively. No significant differences were observed in the age, gender, BMI, diabetes duration, daily insulin dosage, FBG, 2hBG, HbA1c, FCP, 2hCP, or lipid profiles between the two groups (all *p* > 0.05). Moreover, patients in the GV-H group also had a significantly higher standard deviation (SD) of glucose, mean amplitude of glucose excursions (MAGE), and low blood glucose index (LBGI) values than those in the GV-L group in addition to glucose CV (all *p* < 0.001).

### 3.2. Models Analysis in the Exploratory Set

A total of 5597 metabolites were identified (positive mode: 3012; negative mode: 2585). The score plots of the PCA (Figure 1a), PLS-DA (Figure 1b), and OPLS-DA (Figure 1c) models were drawn for serum samples from the GV-H and GV-L groups in the exploratory set. The samples from the GV-H and GV-L groups were separated, indicating that they had significantly different metabolic profiles. The preliminary differential metabolites screening identified 569 differential metabolites from the HMDB and KEGG databases. These metabolites mainly included carboxylic acid and derivatives (20.21%), lipids (20.04%), benzene and substituted derivatives (11.78%), organic compounds (11.42%), purines, pyrimidines and their derivatives (6.68%), a few of the detailed classification unidentified metabolites (3.34%), and other metabolites (Figure 1d).

### 3.3. Pathway Analysis in the Exploratory Set and Candidate GV Biomarkers

A total of 569 differential metabolites identified between the GV-H and GV-L groups were enriched in 82 KEGG metabolic pathways. The top twenty pathways altered between the GV-H and GV-L groups were picked according to the calculated −log10(*p*-value) value as displayed in Figure 2a. Additionally, the results indicated that the top seven pathways were altered significantly (*p* < 0.05, impact > 0.01): linoleic acid metabolism, aminoacyl-tRNA biosynthesis, ATP-binding cassette transporters, taurine and hypotaurine metabolism, phenylalanine metabolism, cysteine and methionine metabolism, and alanine, aspartate, and glutamate metabolism (Appendix A). A total of 25 metabolites were enriched in these seven pathways and were thus considered potential GV biomarkers (Table 2, Figure 2b). 

### 3.4. Candidate GV Biomarkers Validation

Another 21 T1D patients underwent metabolomics analysis to further validate the stability of these 25 differential metabolites (Appendix A). To reduce bias caused by potential confounders, these 21 subjects were divided into 11 with high GV (mean glucose CV = 46.2%) and 10 matched patients with low GV (mean glucose CV = 25.3%). Although phosphatidylcholine, riboflavin, 9,10-DHOME, cysteine-S-sulfate, L-cysteine, hydrocinnamic acid, N-acetyl-L-aspartate, and mannitol were significantly different in the GV-H group compared to the GV-L group in this validation set, their changing trends were opposite to those observed in the exploration set. On the other hand, spermidine, trehalose, and L-methionine were altered similarly to those in the exploratory set (Appendix A). Thus, we finally identified these three differential metabolites mentioned above as potential GV biomarkers.

### 3.5. Performance of Final Selected Biomarkers for Predicting GV

In the exploratory set, spermidine outperformed in distinguishing two groups with an area under the curve (AUC) of 0.879 (95% confidence interval (CI): 0.721–1.000) and sensitivity and specificity of 100.0% and 87.5%, respectively. Trehalose had an AUC of 0.886 (95% CI: 0.772–1.000, sensitivity: 81.3%, specificity: 82.4%), and L-methionine had an AUC of 0.746 (95% CI: 0.577–0.915, sensitivity: 56.3%, specificity: 88.2%), respectively. Furthermore, the AUC of the above three metabolites combined to differentiate the GV-H and GV-L groups was 0.952 (95% CI: 0.887–1.000), with sensitivity and specificity of 100.0% and 81.2%, respectively. In the validation set, the AUCs for spermidine, trehalose, and L-methionine for distinguishing the GV-H and GV-L groups were 0.845 (95% CI: 0.645–1.000, sensitivity: 90.9%, specificity: 80.0%), 0.764 (95% CI: 0.555–0.972, sensitivity: 81.8%, specificity: 60.0%), and 0.764 (95% CI: 0.550–0.977), respectively. The AUC of these three metabolites combined to distinguish the two groups was 0.918 (95% CI: 0.801–1.000), with sensitivity and specificity of 90.9% and 80.0%, respectively (Figure 3).

### 3.6. Correlation between Selected Biomarkers and Glycemic Parameters for All Patients

L-methionine was significantly inversely related to HbA1c (r = −0.427, *p* = 0.001) and FBG (r = −0.329, *p* = 0.017). In terms of CGM parameters, L-methionine and trehalose were inversely related to low blood glucose index (LBGI) (both *p* < 0.05). On the other hand, spermidine was significantly positively correlated, whereas L-methionine was negatively correlated with SD, MAGE, glucose CV, and LBGI (all *p* < 0.01) (Table 3, Figure 4). 

### 3.7. Predictors for GV by Multiple Linear Regression Analysis for All Patients

The correlation between glucose CV and age, BMI, diabetes duration, daily insulin dosage, FBG, HbA1c, FCP, 2hCP, spermidine, L-methionine, and trehalose was analyzed using linear regression analysis by employing a stepwise procedure. The association between glucose CV and BMI (standardized β = −0.288, *p* = 0.036), FCP (standardized β = −0.377, *p* = 0.005), spermidine (standardized β = 0.477, *p* = 0.04), and L-methionine (standardized β = −0.273, *p* = 0.04) reached statistical significance. The R^2^ in this model was 0.446 (F = 8.84, *p* < 0.001).

## 4. Discussion

T1D is an organ-specific autoimmune disease in which pancreatic β-cells suffer varying degrees of immune damage, resulting in absolute insulin deficiency and non-negligible GV [1]. With the increasing use of CGM, several studies have revealed that GV may be involved in the emergence of diabetic complications [17,18,19], although the mechanism is not yet understood. Moreover, GV in some T1D patients remained heterogeneous despite comparable clinical indicators, whether other factors are involved is yet unknown.

In the exploratory set, we identified 569 metabolites that differed between the GV-H and GV-L groups after the primary screening. These metabolites were enriched in 82 metabolic pathways, of which seven were identified as differential metabolic pathways. Subsequently, 25 different metabolites were found to be enriched in these seven pathways. After validation, three candidate GV biomarkers were identified: spermidine, L-methionine, and trehalose. Furthermore, ROC analysis revealed that spermidine, L-methionine, and trehalose adequately distinguished between the GV-H and GV-L groups in both the exploratory and validation sets. In addition to BMI and FCP, multiple linear regression analysis revealed that L-methionine and spermidine were independent predictors of glucose CV.

Spermidine (C_7_H_19_N_3_), such as spermine and putrescine, are natural polyamine compounds [20]. L-arginine is used as a substrate to synthesize polyamines in mammalian cells via arginase in extrahepatic tissues. Polyamines are a class of compounds that contain two or more amino groups and are primarily synthesized from L-arginine and ornithine, with arginine decarboxylase and ornithine decarboxylase serving as key enzymes [21]. Polyamines have been shown to be effective antioxidants that protect essential cellular components in cell membranes, such as polyunsaturated fatty acids in the membrane, from oxidative damage over the last decade [22]. Higher amounts of polyamines have been documented in both exocrine and endocrine pancreatic cells, which may lead to endoplasmic reticulum stress, oxidative stress, inflammatory response, and autophagy [23]. Specifically, polyamines have been shown to play a pathogenic role in T1D progression because they are known to play a critical role in the functioning of β-cells [24] and immune cells [25]. The inhibition of polyamine biosynthesis significantly delays diabetes incidence in NOD mice [26]. Moreover, polyamines have a time- and concentration-dependent inhibitory effect on the protein phosphatase activity of insulin-secreting cells in diabetes [27]. In vivo, spermidine inhibited HbA1c and lipid peroxidation [28]. However, a recent study on the role of spermidine treatment in T1D pathogenesis found that daily oral spermidine treatment in NOD mice resulted in a higher diabetes incidence, organ-specific spermidine accumulation, elevated peripheral inflammation, and a lower proportion of suppressive Tregs [29]. Reportedly, spermidine is linked to diabetic complications. Spermidine levels in the aqueous humor of patients with proliferative diabetic retinopathy (PDR) can be 3–4 times higher than in healthy controls [30]. Furthermore, in insulin-treated diabetic patients, an increase in intra-erythrocytic spermidine content has been linked to both diabetic nephropathy and severe retinopathy [31]. It has also been discovered that children with T1D have higher polyamine oxidase activity, which could lead to increased ROS production and subsequent oxidative damage [32]. Similarly, spermidine oxidase activity, another polyamine catabolic enzyme, is significantly lower in T1D patients compared to non-diabetics [33]. This highlights the fact that decreased activity of polyamine catabolic enzymes in T1D patients may promote polyamine accumulation.

According to our findings, spermidine levels were higher in patients with high GV, and spermidine levels were significantly positively correlated with GV parameters. This phenomenon could be attributed to increased oxidative stress in patients with high GV [34] because oxidative stress caused by GV would be more severe than hyperglycemia alone [33,35,36,37]. The stimulation of oxidative stress by GV is associated with an increase in ROS production, oxidative damage to DNA, and a decrease in superoxide dismutase activity [38]. The current study was the first to show that spermidine levels can be used to predict GV levels in T1D patients, implying that higher spermidine levels are associated with high GV. Thus, combining the current findings with the previously reported tendency of increased spermidine levels in patients with microvascular complications may provide an in-depth insight into the role of GV in the development of diabetic complications. The mechanism of elevated spermidine levels in T1D individuals with high GV is yet to be elucidated. Given that oxidative stress brought on by GV would be more severe than hyperglycemia alone [34,35,36,37,38], one explanation is that spermidine excess is a reactive change brought on by reactive oxygen species (ROS). GV-induced oxidative stress is related to an increase in ROS generation, oxidative DNA damage, and a reduction in superoxide dismutase function [39]. Reportedly, higher ROS generated spermidine, which protected the cell from oxidative damage [40,41]. In this context, changes in spermidine concentration could be a consequence, and may not be the cause, of high GV. So far, however, there are only assumptions regarding the precise mode of action and the impact on GV of different spermidine concentrations. To support or refute any of the foregoing hypotheses, additional in vitro and in vivo mechanistic research is required.

L-methionine (C_5_H_11_NO_2_S) is an essential sulfur-containing α-amino acid that cannot be produced in the body and is only available externally [42]. L-methionine has been shown to inhibit liver gluconeogenesis gene expression by promoting PGC1-acetylation [43]. L-methionine and other methyl donors improve glucose tolerance and insulin sensitivity in mice offspring fed a high-fat diet [44]. Accumulating evidence indicated that L-methionine activates AMPK and SIRT1, a mechanism similar to metformin. Furthermore, L-methionine has been shown to improve the altered key one-carbon metabolite metabolism in T2D rats and diabetes-induced disturbances in glucose and lipid metabolism [45]. These findings suggested that increased methionine levels could be used to treat diseases associated with glucose and lipid metabolism disorders. Patients in the GV-H group had significantly lower serum L-methionine levels than those in the GV-L group in the current study, and L-methionine was inversely correlated with both HbA1c and GV parameters in correlation analysis; thus, L-methionine was an independent predictor of glucose CV, indicating its role as a biomarker of GV in T1D patients. The role of methionine in T1D pathogenesis and progression has also been indicated in several studies. In T1D rats, L-methionine also protected pancreatic β-cells by regulating FOXO1 expression [46]. According to the German BABYDIAB study, children who developed autoantibodies by the age of two had a 2-fold lower methionine content than those who developed autoantibodies later or autoantibody-negative controls [47]. Another ‘Environmental Triggers for Type 1 Diabetes’ (MIDIA) study found that methionine levels in T1D progressors were lower with time, according to longitudinal changes in plasma metabolic profiles [48]. Methionine deficits were also detected in the metabolomics profile of the pre-T1D mice [49]. Furthermore, there is evidence that low methionine levels are associated with increased oxidative stress. Significant T1D-dependent increases in circulating oxidation products and decreases in methionine and cysteine levels were detected, indicating increased oxidative stress [50]. Methionine is a component of the transsulfuration process, which leads to glutathione production, an important intracellular antioxidant. Prior studies have shown that insulin-deprived T1D patients have lower rates of homocysteine-methionine remethylation and higher rates of transsulfuration compared to control subjects [51]. Accordingly, patients with poor glycemic control have lower glutathione pools and lower erythrocyte free cysteine level that can be synthesized from methionine [52]. Therefore, the decrease in circulating methionine may indicate an increased glutathione production. In the present study, patients in the GV-H group had significantly lower serum L-methionine levels than those in the GV-L group, and L-methionine was inversely correlated with HbA1c, FBG, and GV parameters in correlation analysis. Consequently, we speculate that decreases in circulating methionine may serve as a potential biomarker for T1D patients with poor glycemic control and high GV.

Trehalose (C_12_H_22_O_11_) is a non-reduced disaccharide composed of two glucose molecules joined by a hemiacetal hydroxyl group. Despite the fact that trehalose is a non-reduced disaccharide, several in vitro and in vivo studies have confirmed its role as a natural antioxidant [53,54]. Trehalose has antioxidative, anti-inflammatory, and enhanced autophagy functions that can inhibit oxidative stress, inflammation, and autophagy-related diseases, such as diabetes [55,56] and atherosclerosis [57]. Specifically, trehalose’s anti-diabetic effect may be linked to reduced oxidative stress and improved islet function [58]. Furthermore, trehalose induces CD8^+^ Treg cells in mice and preserves insulin concentration, a critical molecule in inhibiting STZ-induced T1D development in mice. Trehalose supplementation completely reverses the mild hyperglycemia (<19.4 mmol/L) in diabetic NOD mice and might also increase CD8^+^ Treg cells in mice that did not respond to treatment. These findings suggested that trehalose could be used to treat T1D when pancreatic β-cell regeneration is possible [59]. However, metabolomics studies revealed that trehalose might be a risk factor for the development of T2D [60,61]. Another metabonomic study on the risk factors for DR in T2D patients showed that low serum trehalose concentrations were associated with DR progression [62]. According to the current findings, serum trehalose levels are higher in T1D patients with low GV. Additionally, the present study was the first to use metabolomics to identify the link between trehalose and GV. It can be seen that uncertainty exists regarding the association between elevated trehalose levels and high GV in T1D patients. The levels of trehalose reflected by the peak intensity in the present study were very low when compared to spermidine and L-methionine. Moreover, it is worth noting that in both the exploratory and validation sets, the FC values of spermidine and L-methionine were relatively consistent. The FC of trehalose in the validation set, however, was less than half that of the FC of trehalose in the exploration set, indicating that the difference of trehalose between the GV-H and GV-L groups may not as reliable as that of spermidine and L-methionine. Therefore, additional in vitro and in vivo mechanistic studies are required to clarify the relationship between high trehalose levels and high GV. 

The present study has some limitations. First, the sample size was small. Although we matched the GV-H and GV-L groups according to age, gender, diabetes duration, BMI, daily insulin dosage, FBG, HbA1c, and CP levels to minimize the confounding factors, the results may be underpowered for the detection of certain biomarkers specific to GV. Second, due to the cross-sectional study design, we could not elucidate the cause-and-effect correlation in this study. Third, future studies are required to test the performance and reliability of selected biomarkers in patients with different levels of GV. One other potential limitation of our study is that we were unable to assess the impact of dietary intake on the circulatory metabolome of T1D patients, which may have contributed to the differences in levels of the present identified metabolites. Future investigation into diet-metabolome interactions will be required to clarify the impact of dietary intake on glycemic control and GV levels of patients with T1D.

## 5. Conclusions

In conclusion, the GV-H and GV-L groups exhibited different metabolic perturbations. Herein, we reported for the first time that increased spermidine and decreased L-methionine and trehalose may contribute to high GV in T1D patients. BMI, FCP, spermidine, and L-methionine were independent predictors of glucose CV. Thus, future studies should focus on determining and validating whether the identified metabolites are related to an increased risk of micro- and macrovascular complications in T1D patients in a prospective, randomly selected population. In conclusion, the current study emphasizes the various metabolic disturbances among T1D patients with different GV levels. We discovered that serum spermidine, L-methionine, and trehalose may impact glycemic control; particularly, high spermidine and low L-methionine levels were associated with high GV. L-methionine was shown to be more closely associated with poor glycemic control and high GV. Differences in trehalose were not as reliable as those in spermidine and L-methionine because of the relatively low amounts of trehalose and the inconsistent fold change sizes in the exploratory and validation sets. These alterations, however, may not be specifically related to GV since dietary intake may also have a partial influence on the metabolomics profiles. Our results also suggest that spermidine and L-methionine, in addition to BMI and FCP, may be useful predictors of glucose CV. Overall, these observations suggest that metabolomic disturbances impact the GV of T1D. Additional in vitro and in vivo mechanistic studies are required to elucidate the relationship between spermidine and L-methionine levels and GV in a cohort of T1D patients with different geographical and nutritional backgrounds.

## Figures and Tables

**Figure 1 nutrients-15-00518-f001:**
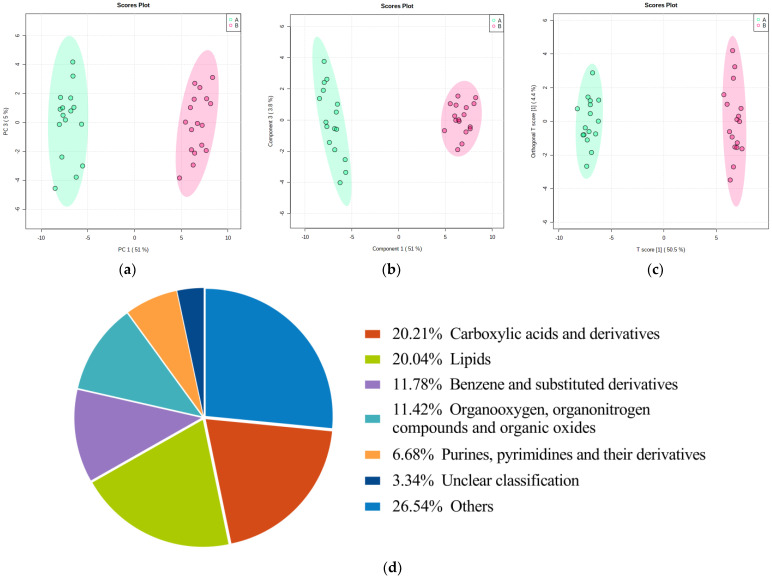
Score plots of the PCA/PLS-DA/OPLS-DA models for discriminating the GV-H group from the GV-L group and pie chart of preliminary differential metabolites screening results in the exploratory set. Samples of these two groups were distinctly separated, indicating that they had markedly different metabolic characteristics (**a**–**c**). Pie chart (**d**) of differentially metabolomics showed that the 569 metabolites mainly involved carboxylic acids and derivatives (20.21%), lipids (20.04%), benzene and substituted derivatives (11.78%), organic compounds (11.42%), purines, pyrimidines, and their derivatives (6.68%), a few of the detailed classification unidentified metabolites (3.34%), and other metabolites.

**Figure 2 nutrients-15-00518-f002:**
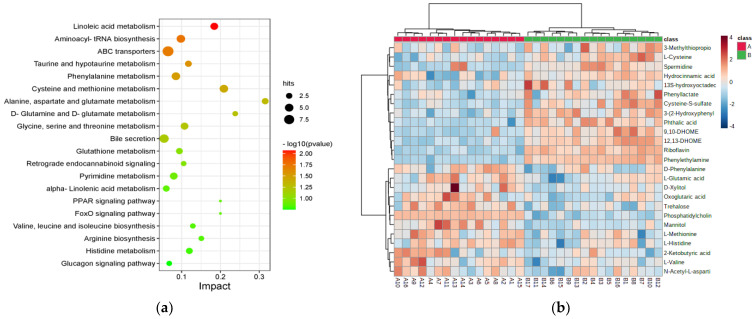
Disturbed metabolic pathways and heatmap of preliminarily selected 25 differential metabolites between the GV-H and GV-L groups in the exploratory set. Bubble diagram of differential metabolic pathways between the GV-H and GV-L groups (**a**). Each point represents a pathway, the abscissa is the Impact value, and the ordinate is the enriched pathway. The dots indicate the number of metabolic molecules corresponding to the pathway. The color is related to the *p* value, the redder the color, the smaller the *p* value. Heatmap of preliminarily selected 25 differential metabolites (**b**). Each row represents a metabolite, and each column depicts a subject. Group A is the GV-L group and group B is the GV-H group; the peak intensity of a metabolite is shown by different colors, namely, the redder the color, the higher the peak intensity.

**Figure 3 nutrients-15-00518-f003:**
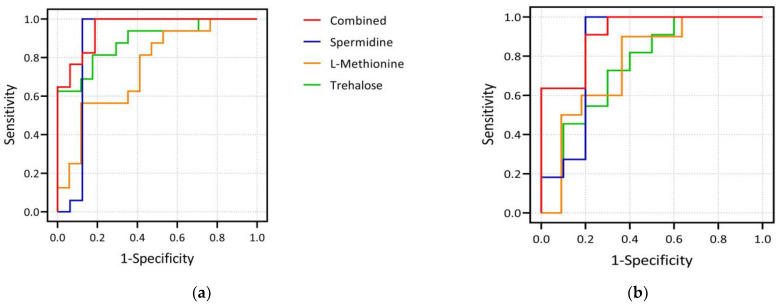
Plots of ROC curve analysis of three differential metabolites. (**a**) shows the ROC curves of spermidine, L-methionine, trehalose and their combination as biomarkers to distinguish GV-H and GV-L groups in the exploration set. The AUCs of spermidine, trehalose, and L-methionine to distinguish the two groups were 0.879, 0.886, and 0.746, respectively, and the AUC of the combination of the three indicators to distinguish the two groups was 0.952. (**b**) shows the corresponding results in the validation set. The AUCs of spermidine, trehalose, and L-methionine to distinguish the two groups were 0.845, 0.764, and 0.764, respectively, and the AUC of the combination of the three was 0.918. It can be seen that the three selected biomarkers in our study, whether used alone or in combination, performed well in distinguishing the GV-H and GV-L groups.

**Figure 4 nutrients-15-00518-f004:**
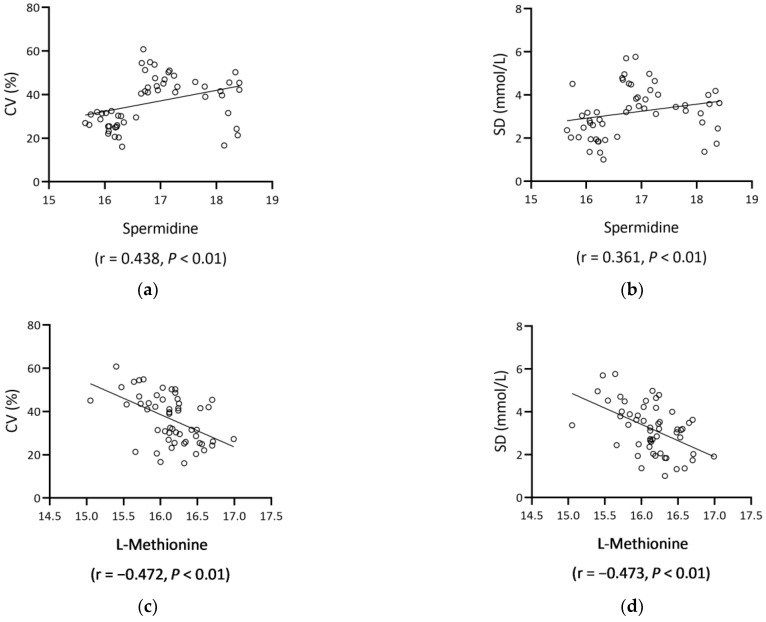
Scatter plots of spermidine/L-methionine and GV metrics (glucose CV, SD). Spermidine positively correlated with glucose CV ((**a**), r = 0.438, *p* < 0.01) and SD ((**b**), r = 0.361, *p* < 0.01), and L-methionine negatively correlated with glucose CV ((**c**), r = −0.472, *p* < 0.01) and SD ((**d**), r = −0.473, *p* < 0.01). Abbreviations: CV, coefficients of variation; SD, standard deviation.

**Table 1 nutrients-15-00518-t001:** Basic characteristics and CGM parameters in GV-H and GV-L groups of the exploratory set.

Characteristics	GV-H Group (*n* = 17)	GV-L Group (*n* = 16)	*p* Value
Age, years, median [Q1, Q3]	18.0 (12.0, 28.0)	18.0 (14.0, 32.5)	0.691
Sex, M/F, *n*	10/7	9/7	0.883
BMI, kg/m^2^, mean (SD)	18.8 ± 3.2	20.1 ± 2.5	0.182
Duration, years, median [Q1, Q3]	1.8 (1.0, 3.7)	1.5 (0.7, 5.4)	0.732
Insulin, U/kg·day, median [Q1, Q3]	0.67 (0.56, 0.82)	0.63 (0.45, 0.74)	0.165
FBG, mmol/L, median [Q1, Q3]	8.4 (7.5, 9.1)	7.7 (6.3, 11.3)	0.634
2hBG, mmol/L, median [Q1, Q3]	17.3 (15.7, 19.6)	16.2 (13.4, 17.8)	0.239
HbA1c, %, mean (SD)	8.4 ± 1.8	8.4 ± 2.1	1.000
FCP, pmol/L, median [Q1, Q3]	59.8 (20.9, 91.2)	42.9 (21.8, 130.2)	0.600
2hCP, pmol/L, median [Q1, Q3]	129.3 (63.9, 176.0)	66.1 (30.6, 227.7)	0.704
TC, mmol/L, mean (SD)	4.1 ± 0.7	4.2 ± 0.8	0.704
TG, mmol/L, median [Q1, Q3]	0.55 (0.49, 0.72)	0.70 (0.63, 1.01)	0.118
HDL, mmol/L, mean (SD)	1.5 ± 0.3	1.4 ± 0.5	0.903
LDL, mmol/L, mean (SD)	2.3 ± 0.6	2.5 ± 0.6	0.613
Glucose SD, mmol/L, median [Q1, Q3]	3.6 (3.4, 4.5)	2.6 (2.0, 3.1)	<0.001
MAGE, mmol/L, median [Q1, Q3]	8.5 (7.6, 9.6)	5.4 (4.2, 7.0)	<0.001
Glucose CV, %, mean (SD)	46.3 ± 5.7	26.8 ± 3.7	<0.001
LBGI, median [Q1, Q3]	5.2 (4.2, 6.9)	1.3 (0.4, 1.9)	<0.001

Note: Data were expressed as mean ± SD, median (25th percentile, 75th percentile) and ratio. BMI, body mass index; FBG, fasting blood glucose; 2hBG, 2 h postprandial blood glucose; HbA1c, hemoglobin A1c; FCP, fasting C-peptide; 2hCP, 2 h postprandial C-peptide; TC, total cholesterol; HDL, high-density lipoprotein; LDL, low-density lipoprotein; TG, triglyceride; SD, standard deviation; MAGE, mean amplitude of glucose excursions; CV, coefficient of variation; LBGI, low blood glucose index.

**Table 2 nutrients-15-00518-t002:** Candidate novel GV biomarkers identified in seven differential metabolic pathways.

Metabolites	KEGG	FC	*p* Value	VIP	Trend
Phosphatidylcholine	C00157	119.49	1.71 × 10^−9^	1.30	Increased
Trehalose	C01083	3.33	0.000056	1.66	Increased
Mannitol	C00392	2.92	4.62 × 10^−6^	1.80	Increased
D-Xylitol	C00379	2.55	0.005381	1.13	Increased
Oxoglutaric acid	C00026	1.56	0.013587	1.27	Increased
L-Methionine	C00073	1.44	0.015133	1.33	Increased
D-Phenylalanine	C02265	1.31	0.033795	1.05	Increased
L-Valine	C00183	1.3	0.002162	1.41	Increased
L-Histidine	C00135	1.29	0.003685	1.32	Increased
L-Glutamic acid	C00025	1.28	0.022906	1.11	Increased
2-Ketobutyric acid	C00109	1.28	0.040674	1.08	Increased
N-Acetyl-L-aspartic acid	C01042	1.21	0.037103	1.11	Increased
Spermidine	C00315	0.56	0.000083	1.49	Decreased
13S-hydroxyoctadecadienoic acid	C14762	0.53	0.007721	1.34	Decreased
Hydrocinnamic acid	C05629	0.53	0.013587	1.32	Decreased
Riboflavin	C00255	0.51	1.71 × 10^−9^	1.31	Decreased
Phenyllactate	C05607	0.48	0.001224	1.29	Decreased
3-(2-Hydroxyphenyl)propanoic acid	C01198	0.47	6.39 × 10^−7^	1.05	Decreased
3-Methylthiopropionic acid	C08276	0.43	0.033795	1.01	Decreased
L-Cysteine	C00097	0.23	0.000208	1.67	Decreased
9,10-DHOME	C14828	0.22	1.17 × 10^−6^	2.06	Decreased
Cysteine-S-sulfate	C05824	0.16	0.000012	1.84	Decreased
Phthalic acid	C01606	0.12	1.71 × 10^−9^	1.18	Decreased
Phenylethylamine	C05332	0.02	1.71 × 10^−9^	2.56	Decreased
12,13-DHOME	C14829	0.02	1.71 × 10^−9^	2.36	Decreased

Note: FC, fold change (the peak intensity ratio of target metabolite in the GV-L and GV-H groups); *p* value, *p* value of hypergeometric distribution test (the smaller *p* value, the more significant the effect of detected differential metabolites on this pathway); VIP, variable importance in projection (a measure of the intensity and explanatory ability of the effect of different metabolite accumulation on the classification and discrimination of samples in each group; VIP ≥ 1 is a common differential metabolite screening criteria); Trend, FC > 1 indicates that the metabolite is down-regulated in the GV-H group, FC < 1 indicates that the metabolite is increased in the GV-H group compared with the GV-L group in the exploratory set. 9,10-DHOME, (12Z)-9,10-Dihydroxyoctadec-12-enoic acid; 12,13-DHOME, (9Z)-12,13-Dihydroxyoctadec-9-enoic acid.

**Table 3 nutrients-15-00518-t003:** Correlation of selected biomarkers with glycemic parameters in all subjects (r).

	Spermidine	L-Methionine	Trehalose
HbA1c	0.105	−0.427 **	0.197
FBG	−0.076	−0.329 *	0.155
Glucose SD	0.361 **	−0.473 **	−0.117
MAGE	0.351 **	−0.385 **	−0.102
Glucose CV	0.438 **	−0.472 **	−0.261
LBGI	0.367 **	−0.279 *	−0.419 **

Note: Values in the table represent Spearman’s correlation coefficient. HbA1c, hemoglobin A1c; FBG, fasting blood glucose; SD, standard deviation MAGE, mean amplitude of glucose excursions; CV, coefficient of variation; LBGI, low blood glucose index. * *p* < 0.05, ** *p* < 0.01.

## Data Availability

All relevant data are within the manuscript and its Appendix A.

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
