# Peer review of "Serum Metabolomics Reveals a Potential Benefit of Methionine in Type 1 Diabetes Patients with Poor Glycemic Control and High Glycemic Variability"

_nutrients, 2023, doi:10.3390/nu15030518_

Round 1

Reviewer 1 Report

The authors present a biomarker discovery for T1D patients with high glycemic variability using untargeted metabolomics approach, which would be of great interest to a broad readership. The strategy from discovery to validation is sound, where the conclusion is carefully drawn, and the discussion is interesting and informatics based on published results. However, there are several concerns which need to be addressed before acceptance.

Language editing is needed for the whole manuscript. e.g. for metabolites, instead of using up/down regulated, mostly using increase/decrease; Line 12 "whether" should be "and whether" because they are two sentences; Line 15 "done on 17 patients" should be "done on serum samples from 17 patients..."

In the abstract, before introducing the metabolomics profiling, a short sentence should be added on why using metabolomics approach. 

Full name for acronyms when first present in the manuscript, e.g. HbA1c and KEGG in abstract.

A lot of details are missing in the method section, e.g. methods and references for testing HbA1c levels and CP levels; Although you referenced two papers for the metabolomics analysis, procedures should still be briefly wrote out, for example, which specific LC-MS systems, what mobile phases, gradients, columns, how samples were extracted, how MS system was calibrated, how metabolites were identified? etc.

what does LOESS stand for?

There is a big concern on the metabolite identification: do you have standard library for retention time references or it is mostly based on MS2 fragmentation patterns? because the mass accuracy was not outstanding (30ppm). And the phrase of "metabolite concentrations was obtained by comparing the mean peak intensities" in Line 137 cannot be accepted, because peak intensity doesn't directly reflect concentrations, and this is only relative quantitation. How many metabolites in total were identified and what classes do they belong to? This information should be present before the PLS-DA results in section 3.2. How did serum sample contain "benzene and substituted derivatives"? How was the metabolites classified? There is a separated class called organic compounds, but shouldn't all metabolites be organic compounds? What does it mean by "classified unidentified metabolites"? 

Fig 2A is showing more than 7 pathways. What are they and how were they picked?

PCA was mentioned in method section, why its results not shown?

Minor modifications:

Although people will know which CV you are talking about after reading the whole paper, it is suggested to say "glucose CV" in the abstract instead of "CV".

Full name for 12,13-DHOME.

Line 131-132, the full names for PLS and OPLS are noted wrong.

"1-specificity" should be "specificity".

Line 278 "standardized =" should be "standardized β = "

Missing reference for Line 327-328.

Author Response

Response to Reviewer 1 Comments

The authors present a biomarker discovery for T1D patients with high glycemic variability using untargeted metabolomics approach, which would be of great interest to a broad readership. The strategy from discovery to validation is sound, where the conclusion is carefully drawn, and the discussion is interesting and informatics based on published results. However, there are several concerns which need to be addressed before acceptance.

Thank you very much for your summary and comments. We appreciate your insightful comments and meticulous revisions to our manuscript. We gained a deeper understanding of metabolomics and harvested a lot of professional knowledge by addressing your concerns one by one. Our manuscript has been significantly improved as a result of your professional comments. We carefully considered your suggestions, revised the manuscript, and provided detailed explanations below. Please do not hesitate to contact us if there are any additional revisions we can make. We sincerely hope that this revised version is of higher quality and more appropriate for publication. All changes made in the paper have been highlighted in red font to aid further evaluation of our revised manuscript.

Point 1: Language editing is needed for the whole manuscript. e.g. for metabolites, instead of using up/down regulated, mostly using increase/decrease; Line 12 "whether" should be "and whether" because they are two sentences; Line 15 "done on 17 patients" should be "done on serum samples from 17 patients..."

Response 1: Thanks for your insightful suggestion. To address your concern, the whole manuscript has undergone through language editing (we made corresponding changes directly in the revised manuscript instead of marking them out in red as there are many small modifications). We have used increase/decrease for metabolites’ changes. Line 12 “whether” has been changed to “and whether”. Line 15 “done on 17 patients” has been changed to “carried out on serum samples from 17 patients...”. Please refer to our revised manuscript.

Point 2: In the abstract, before introducing the metabolomics profiling, a short sentence should be added on why using metabolomics approach.

Response 2: Thanks for your suggestion. To address your concern, we have added the following sentence in the Abstract: “Metabolites in the serum indicate a broad effect of GV on cellular metabolism and therefore are more likely to indicate metabolic dysregulation associated with T1D.” (Line 13-14)

Point 3: Full name for acronyms when first present in the manuscript, e.g. HbA1c and KEGG in abstract.

Response 3: Thanks for reminding us about this. We have added the full name for acronyms of BMI, HbA1c and KEGG where they first appeared.

Point 4: A lot of details are missing in the method section, e.g. methods and references for testing HbA1c levels and CP levels; Although you referenced two papers for the metabolomics analysis, procedures should still be briefly wrote out, for example, which specific LC-MS systems, what mobile phases, gradients, columns, how samples were extracted, how MS system was calibrated, how metabolites were identified? etc.

Response 4: Thanks for this insightful comment. We apologize for our rough description of this important information. For testing HbA1c and the CP levels, we have added relevant information in the text: “Automated high-performance liquid chromatography was used to measure HbA1c levels (VARIANT II Haemoglobin Testing System; Bio-Rad Laboratories), which is the gold standard method for monitoring glucose control in diabetes patients as recom-mended by the American Diabetes Association (ADA) [12]. The serum CP levels were de-termined using a chemiluminescent method and an Adivia Centaur XP immunoassay system (Siemens, Germany). The inter-assay and intra-assay variation coefficients of CP test were 3.7–4.1% and 1.0–3.3%, respectively.” (Line 110-116)

Then, we also add the detailed procedures of sample preparation, metabolomic analysis and metabolite identification in the text: “First, the sample was thawed at 4 °C, vortexed for 1 min, and mixed evenly. Second, an appropriate amount of the sample was transferred into a 2 mL centrifuge tube. Third, 400 µL methanol (stored at -20 °C) was added and vortexed for 1 min. Then, the supernatant was collected by centrifugation of the sample at 12,000 rpm, 4 °C for 10 min; it was further concentrated and dried. Finally, 150 µL of 2-chloro-l-phenylalanine (4 ppm) solution pre-pared with 80% methanol water (stored at 4 °C) was added to resolubilize the sample, and the supernatant was filtered through a 0.22 μm membrane and transferred into the detec-tion bottle for LC-MS detection.

LC analysis was performed on a Vanquish UHPLC System (Thermo Fisher Scientific, USA). The separation was conducted under the following gradient: 0-1 min, 2% C; 1–9 min, 2–50% C; 9–12 min, 50–98% C; 12–13.5 min, 98% C; 13.5-14 min, 98–2% C; 14–20 min, 2% C. For LC-ESI (-)-MS analysis, the analytes were carried out with (A) acetonitrile and (B) ammonium formate (5 mM). Chromatography was performed on a Waters AC-QUITY UPLC® HSS T3 (150 × 2.1 mm, 1.8 µm) column maintained at 40 °C. The injection volume and flow rate were set at 0.25 mL/min and 2 L, respectively. The mobile phases for LC-ESI (+)-MS analysis were (C) 0.1% formic acid in acetonitrile (v/v) and (D) 0.1% formic acid in water (v/v). The gradient was as follows: 0-1 min, 2% C; 1-9 min, 2-50% C; 9-12 min, 50-98% C; 12-13.5 min, 98% C; 13.5-14 min, 98-2% C; 14-20 min, 2% C. The analytes for LC-ESI (-)-MS analysis were (A) acetonitrile and (B) ammonium formate (5 mM). The gra-dient used for the separation was as follows: 2% A for 0–1 min; 2–50 min; 50–98 min; 12–13.5 min; 98% A; 13.5–14 min; 98–2% A; and 2% A for 1–17 min. Metabolites were detect-ed by mass spectrometry on an Orbitrap Exploris 120 (Thermo Fisher Scientific) with an ESI ion source. The simultaneous acquisition of MS1 and MS/MS (full MS-ddMS2 mode, data-dependent MS/MS) was employed. The parameters were as follows: sheath gas pressure, 30 arb; aux gas flow, 10 arb; spray voltage, 3.50 kV and -2.50 kV for ESI (+) and ESI (-), respectively; capillary temperature, 325 °C; MS1 range, m/z 100–1000; MS1 resolv-ing power, 60000 FWHM; number of data-dependent scans per cycle, 4; MS/MS resolving power, 15000 FWHM; normalized collision energy, 30%; dynamic exclusion time, auto-matic.

To correct for any systematic bias, the robust locally estimated scatterplot smoothing (LOESS) signal correction (QC-RLSC) was used for data normalization [15]. To guarantee accurate metabolite identification, only ion peaks with relative standard deviations (RSDs) 30% in quality control (QC) were maintained following normalization. The raw UPLC Q-TOF MS data were first converted to mzXML format using MSConvert in the Prote-oWizard software program (v3.0.8789) before being processed with XCMS for feature de-tection, retention time correction and alignment, peak discrimination, filtering, alignment, matching, and identification. Then, a peak table with mz value, retention time (rt), and the corresponding peak intensity was generated. The metabolites were identified by accuracy mass (< 30 ppm) and MS/MS data that were compared to multiple databases. Metabolite identification was first confirmed based on the exact molecular weight and subsequently on the Human Metabolome Database (HMDB) according to the MS/MS fragmentation patterns (http://www.hmdb.ca), massbank (http://www.massbank.jp/), LipidMaps (http://www.lipidmaps.org), mzcloud (https://www.mzcloud.org), and Kyoto Encyclope-dia of Genes and Genomes (KEGG) databases (https://www.genome.jp/kegg/). Finally, the metabolites were validated by comparing retention time and fragmentation pattern to au-thentic standards. Differential metabolites were sought from the sample primary sub-stance list and screened at the preset P-value and variable importance on projection (VIP) thresholds in the statistical test. ” (Line 131-179)

Point 5: what does LOESS stand for?

Response 5: Thanks for this comment. LOESS refers to a univariate approach termed quality control–based robust LOESS (locally estimated scatterplot smoothing) signal correction (QC-RLSC). The drift in the signal for each metabolic feature in each subject sample is corrected by observing the change in signal for the same metabolic feature in ‘bracketing’ QC samples and correcting in the subject samples for the temporal shift in signal related to that observed for QC samples. This is essential in metabolic profiling experiments. In methods providing absolute quantification using calibration curves, changes in response are ‘normalized’ in the calibration process. This is not available in metabolic profiling and, therefore, a separate method for signal correction of data to remove biases in response related to analysis order is required. The method provides data for each block that can be easily integrated.

Point 6: There is a big concern on the metabolite identification: do you have standard library for retention time references or it is mostly based on MS2 fragmentation patterns? because the mass accuracy was not outstanding (30ppm). And the phrase of "metabolite concentrations was obtained by comparing the mean peak intensities" in Line 137 cannot be accepted, because peak intensity doesn't directly reflect concentrations, and this is only relative quantitation. How many metabolites in total were identified and what classes do they belong to? This information should be present before the PLS-DA results in section 3.2. How did serum sample contain "benzene and substituted derivatives"? How was the metabolites classified? There is a separated class called organic compounds, but shouldn't all metabolites be organic compounds? What does it mean by "classified unidentified metabolites"?

Response 6: Thanks for your insightful comment. As for metabolite identification, the raw UPLC Q-TOF MS data were firstly converted to mzXML format by MSConvert in ProteoWizard software package (v3.0.8789) and processed using XCMS for feature detection, retention time correction and alignment, peak discrimination, filtering, alignment, matching, and identification. The metabolites were identified by accuracy mass (< 30 ppm) and MS/MS data which were matched with several databases. Metabolite identification was first confirmed on the basis of exact molecular weight and subsequently on the HMDB based on MS/MS fragmentation patterns, massbank, LipidMaps, mzclound as well as KEGG databases. Finally, metabolites were confirmed by comparison of retention time and fragmentation pattern with authentic standards. Herein, the retention time and MS2 fragmentation patterns were both considered in the metabolite identification.

 We apologize for our inappropriate description regarding “The fold change (FC) in metabolite concentrations was obtained by comparing the mean peak intensities…”. It is true as you pointed out that the peak intensity did not directly reflect concentrations but only relative quantitation. Therefore, we have changed this sentence to: “The fold change (FC) was determined according to the relative quantification reflected by the peak intensity of the metabolite between two groups.”

 There were 5,597 metabolites in total identified (positive mode: 3,012; negative mode: 2,585). They were drugs, steroids and steroid derivatives, pyrimidine nucleotides, purine nucleosides, prenol lipids, organooxygen compounds, indoles and derivatives, flavonoids, fatty acyls, carboxylic acids and derivatives, benzene and substituted derivatives and others. And we have added this to the Result.

 Metabolites were classified based on their chemical structure. For instance, aspirin, benzamide and phthalic acid were all categorized into benzene and substituted derivatives since they all contain phenyl ring structures. And we apologize for our carelessness regarding the classification of “Organic compounds”. Actually, metabolites of this classification include “Organooxygen compounds”, “Organonitrogen compounds” and “Organic oxides”. To make it more accurate, we have corrected this classification name to Organooxygen, organonitrogen compounds and organic oxides. There is a wrong spell in “classified unidentified metabolites”, that is, “classification” was misspelled as “classified”. So, this classification is “classification unidentified metabolites”, which refers to those without a detailed class.

Point 7: Fig 2A is showing more than 7 pathways. What are they and how were they picked?

Response 7: Thanks for your comment. Fig 2A shows the top 20 pathways altered between the high and low glycemic variability groups. We picked 20 pathways from top to bottom according to the calculated -log10(p-value) size for drawing Fig 2A in order to inform the reader which pathway abnormalities were present. And the results indicated that the top 7 pathways were altered significantly (p-value < 0.05): linoleic acid metabolism, aminoacyl-tRNA biosynthesis, ATP-binding cassette transporters, taurine and hypotaurine metabolism, phenylalanine metabolism, cysteine and methionine metabolism, and alanine, aspartate, and glutamate metabolism.

Point 8: PCA was mentioned in method section, why its results not shown?

Response 8: Thanks for your comment. The PCA result has already been added to Figure 1.

Minor modifications:

Point 9: Although people will know which CV you are talking about after reading the whole paper, it is suggested to say "glucose CV" in the abstract instead of "CV".

Response 9: Thanks for your suggestion. We have changed “CV” to “glucose CV” in the text.

Point 10: Full name for 12,13-DHOME.

Response 10: Thanks for your comment. The full name for 12,13-DHOME is (9Z)-12,13-Dihydroxyoctadec-9-enoic acid. The full name of 9,10-DHOME is (12Z)-9,10-Dihydroxyoctadec-12-enoic acid. And they are both indicated in the text now.

Point 11: Line 131-132, the full names for PLS and OPLS are noted wrong.

Response 11: Thanks for your careful check. We have corrected this mistake.

Point 12: "1-specificity" should be "specificity".

Response 12: Thanks for your careful check. We are not sure if this point refers to the ROC curve (Fig 3). If yes, we think the abscissa is “1-specificity”.

Point 13: Line 278 "standardized =" should be "standardized β = "

Response 13: Thanks for your careful check. We have corrected this in the text.

Point 14: Missing reference for Line 327-328

Response 14: Thanks for your careful check. We have added the relevant reference in the text.

We show special gratitude for all your comments.

Reviewer 2 Report

Nutrients_2100038_Serum metabolomics reveals a potential benefit of methionine in type 1 diabetes patients with poor glycemic control and high glycemic variability

This manuscript addresses an important issue that is still poorly understood. The study has a reasonable sample size and appropriate methodology. The results however indicate that there is large variability between the two cohorts which puts the reliability of the results into question. I have some detailed comments below.

1.     Table 1, I do not think that the LBGI can be in mmol/L, shouldn’t this be %? Also, please make clear that the SD is for glucose. It might be better to have all glucose measures together in the table.

2.     Table 2: I am surprised that one of the top metabolites identified is trehalose which is a sugar compound not present in humans but common in bacteria, funghi, plants and insects. Humans cannot actually absorb this sugar. This could suggest that some of the metabolites identified are contaminations or that the settings for metabolite recognition are set incorrectly.

3.     Supplementary tables: I cannot see the supplementary tables which is problematic given that the results from the metabolomics analysis vary so widely between the discovery and validations cohorts. Some of the compounds with the highest fold changes and therefore p-values such as phosphatidylcholine and riboflavin change direction which puts the full set of results into question. Why is the difference so large between the cohorts and how reliable are the remaining results (e.g. those that change in the same direction)? How generalisable are the results if the authors cannot replicate the majority even in a cohort of relatively similar individuals living in the same location and eating similar food?

4.     Figure 3: given that the results vary so widely between the discovery and validation cohorts and that the authors only chose the 3 metabolites that were similarly changed in both cohorts, the performance of the final selected biomarkers is good. However, it would be far more valuable if the performance of these three metabolites were checked in a totally independent cohort. This needs to be done in order to ascertain the reliability of the results.

5.     Table 3: is the negative correlation between methionine (and the positive correlation between spermidine) and the glucose variability parameters describing the relationship between a diet higher in animal protein intake versus not? What information on dietary intake (a major determinant of circulating metabolites) do the authors have and have they done this analysis? Is the lack of correlations between trehalose and glucose variability parameters indicating that the levels of trehalose are very small or is there another explanation?

6.     Paragraph 3.7: from the description, it is not clear how the authors have done the stepwise linear regression analysis. Many of the parameters used here for glucose variability would be co-linear, has collinearity been taken into account? Please explain your process of linear regression better.

Author Response

Response to Reviewer 2 Comments

Nutrients_2100038_Serum metabolomics reveals a potential benefit of methionine in type 1 diabetes patients with poor glycemic control and high glycemic variability

This manuscript addresses an important issue that is still poorly understood. The study has a reasonable sample size and appropriate methodology. The results however indicate that there is large variability between the two cohorts which puts the reliability of the results into question. I have some detailed comments below.

Thank you very much for your summary and comments. We appreciate your insightful comments and meticulous revisions to our manuscript. We gained a deeper understanding of metabolomics and harvested a lot of professional knowledge by addressing your concerns one by one. Our manuscript has been significantly improved as a result of your professional comments. We carefully considered your suggestions, revised the manuscript, and provided detailed explanations below. Please do not hesitate to contact us if there are any additional revisions we can make. We sincerely hope that this revised version is of higher quality and more appropriate for publication. All changes made in the paper have been highlighted in red font to aid further evaluation of our revised manuscript.

The whole manuscript has undergone thorough language editing (we made corresponding changes directly in the revised manuscript instead of marking them out in red as there are many small modifications).

Point 1: Table 1, I do not think that the LBGI can be in mmol/L, shouldn’t this be %? Also, please make clear that the SD is for glucose. It might be better to have all glucose measures together in the table.

Response 1: Thanks for your comment. To address your concern, we referred to relevant literature about LBGI and confirmed that LBGI is a composite score proposed by Kovatchev at the 1,990s (Kovatchev BP. Diabetes Care. 1998;21(11):1870-5). This measure can be used to predict the risk of severe hypoglycemia in the preceding 3 to 6 months. The LBGI is a unitless metric, and people can roughly infer the hypoglycemia risk by the LBGI value (<2.5, low risk; 2.5-5, medium risk; >5, high risk). Herein, we have deleted the “mmol/L” after LBGI in our text.

To make it more accurate, we have changed “SD” to “Glucose SD” in Table 1. For glycemic variability parameters, we only used the standard deviation (SD) of glucose, mean amplitude of glucose excursions (MAGE), coefficient of variation of glucose and the low blood glucose index (LBGI). So we only included these measures in the table.

Point 2: Table 2: I am surprised that one of the top metabolites identified is trehalose which is a sugar compound not present in humans but common in bacteria, funghi, plants and insects. Humans cannot actually absorb this sugar. This could suggest that some of the metabolites identified are contaminations or that the settings for metabolite recognition are set incorrectly.

Response 2: Thanks for your comment. Actually, as we depicted in our manuscript, we first determined 569 differential metabolites, which were enriched in 82 KEGG metabolic pathways, with a total of 7 significantly altered metabolic pathways. Then we further explored the corresponding 25 metabolites identified in these 7 pathways. And trehalose is one of these 25 metabolites (identified from the ABC transporters pathway). Trehalose is a disaccharide of two glucose molecules, which is added to food and other manufactured products to prevent dehydration and protein denaturation. Trehalose is also regarded as a kind of carbohydrate and energy-related metabolite. Two relevant plasma and serum metabolomics profiling studies in patients with diabetes suggested that trehalose could be detected and that higher trehalose levels were closely related to their study outcomes (1. Morze J. Diabetes Care. 2022, 1;45(4):1013-1024; 2. Rebholz CM. Diabetologia. 2018;61(5):1046-1054).

Trehalose can be hydrolyzed by trehalase, a glucosidase enzyme that has a specific effect on trehalose and becomes 2 molecules of glucose after hydrolysis, EC 3.2.1.28. It is widespread in bacteria, molds, plants, and animals. In humans, in addition to the kidneys, intestines, liver, biliary fluid, and urine, its activity can also be detected in the plasma α2-globulin fraction, particularly in the kidneys. We speculate that trehalose content may come from diet or, as you say, due to contamination. We acknowledge that this is one of our limitations. 

Point 3: Supplementary tables: I cannot see the supplementary tables which is problematic given that the results from the metabolomics analysis vary so widely between the discovery and validations cohorts. Some of the compounds with the highest fold changes and therefore p-values such as phosphatidylcholine and riboflavin change direction which puts the full set of results into question. Why is the difference so large between the cohorts and how reliable are the remaining results (e.g. those that change in the same direction)? How generalisable are the results if the authors cannot replicate the majority even in a cohort of relatively similar individuals living in the same location and eating similar food?

Response 3: Thanks for your comment. We apologize that we failed to upload the supplementary tables to the system. As Table S3 showed that although phosphatidylcholine, riboflavin, 9,10-DHOME, cysteine-S-sulfate, L-cysteine, hydrocinnamic acid, N-acetyl-L-aspartate, and mannitol were significantly different in the GV-H group compared to the GV-L group in the validation set, their changing trends were opposite to those observed in the exploration set. However, spermidine, trehalose, and L-methionine, on the other hand, changed in the same way as the exploratory set. As a result, we finally identified these three differential metabolites as potential glycemic variability biomarkers. The reason why is the difference so large between the exploratory and validation cohorts may due to the individual differences. Since our study is only a cross-sectional study, our results only indicated potential relationships between certain metabolites and glycemic control as well as variability. Herein, the conclusion we draw is limited to our study population as we added in the abstract.

 Moreover, we are planning to conduct a prospective study to further illustrate the relationship between serum metabolomics profile and glycemic control in a larger sample size consisting of patients with type 2 diabetes in order to test the generalizability of the present study results.

Point 4: Figure 3: given that the results vary so widely between the discovery and validation cohorts and that the authors only chose the 3 metabolites that were similarly changed in both cohorts, the performance of the final selected biomarkers is good. However, it would be far more valuable if the performance of these three metabolites were checked in a totally independent cohort. This needs to be done in order to ascertain the reliability of the results.

Response 4: Thanks for your comment. As we replied to the 3rd comment, we are planning to conduct a prospective study to further illustrate the relationship between serum metabolomics profile and glycemic control in a larger sample size consisting of patients with type 2 diabetes in order to test the generalizability of the present study results. Furthermore, we are also planning to enroll more than 120 T1D patients with different glycemic variability and tested the performance of selected biomarkers. We have written this in the discussion part as our next step. However, this external validation cannot be done at present because the COVID-19 pandemic hinders our eligible participant’s enrollment. We make sure that this will be done in the future.

Point 5: Table 3: is the negative correlation between methionine (and the positive correlation between spermidine) and the glucose variability parameters describing the relationship between a diet higher in animal protein intake versus not? What information on dietary intake (a major determinant of circulating metabolites) do the authors have and have they done this analysis? Is the lack of correlations between trehalose and glucose variability parameters indicating that the levels of trehalose are very small or is there another explanation?

Response 5: Thanks for your comment. As for food intake during CGM wearing, we have asked the patients to write down their eating times, food categories, and rough evaluations about the approximate amount of food. Patients were requested to maintain their eating habits as usual in order to draw a real glycemic pattern. We looked into the dietary log of patients before the analysis was conducted. But we found it hard to calculate and compare the dietary intake profiles among patients since the patient's diet was not accurately recorded in grams, and we were unable to determine the authenticity of the patient's log. In addition, since our findings were based on the patient's original eating habits that our conclusions were to some extent most consistent with the patient's real situation. Therefore, this has some guiding significance for their diet.

 The lack of correlations between trehalose and glucose variability parameters indicated that the levels of trehalose were very small, and the contribution of trehalose on glycemic variability was not as large as that of the spermidine and L-methionine. This was also confirmed by the following stepwise linear regression analysis, which also indicated the importance of spermidine and L-methionine on glycemic variability.

Point 6: Paragraph 3.7: from the description, it is not clear how the authors have done the stepwise linear regression analysis. Many of the parameters used here for glucose variability would be co-linear, has collinearity been taken into account? Please explain your process of linear regression better.

Response 6: Thanks for your comment. As we originally noted that a number of clinical factors may affect GV, such as diabetes duration, insulin dose, HbA1c levels, and C-peptide levels. Herein, we aimed to explore the relationship of glucose CV with age, BMI, duration of diabetes, FBG, HbA1c, daily insulin dosage, FCP, 2hCP and our selected potential biomarkers (spermidine, L-methionine, and trehalose) were analyzed using linear regression analysis by employing a stepwise procedure. First, we selected the glucose CV as the dependent variable, then the factors aforementioned were regarded as independent variables. The method was set as stepwise regression because it proved to be efficient in picking independent variables that are really useful for predicting CV. We acknowledged that many of the parameters used here for glucose variability would be co-linear, we have taken collinearity into consideration because the stepwise regression model we chose automatically eliminates the collinearity overstatement. Herein, we eventually draw the conclusion that the association between glucose CV and BMI, FCP, spermidine and L-methionine reached statistical significance.

We show special gratitude for all your comments.

Reviewer 3 Report

I do not have access to the supplementary tables. Please provide this.

The authors describe a novel and important evaluation of blood metabolomic characteristics associated with high glycemic variability in type-1 diabetic patients. The sample size is small but the study design is sound and the results are reported and interpreted clearly.

Comments:

Line 48, 286: sentences need reconstruction.

56: "...is complicated." In this and the preceding paragraph, the present tense is appropriate. An exception would be a statement such as, "Smith et al. reported..."

Study participants: How many patients were approached for participation and how many declined? A flow chart of participant assignment in the study would be helpful. If available, report the characteristics of patients enrolled vs. not enrolled and evaluate for differences.

Data collection: The collection of anthropomorphic and blood pressure measurements should be described.

Author Response

Response to Reviewer 3 Comments

I do not have access to the supplementary tables. Please provide this.

The authors describe a novel and important evaluation of blood metabolomic characteristics associated with high glycemic variability in type-1 diabetic patients. The sample size is small but the study design is sound and the results are reported and interpreted clearly.

Thank you very much for your summary and comments. We appreciate your insightful comments and meticulous revisions to our manuscript. Our manuscript has been significantly improved as a result of your professional comments. We carefully considered your suggestions, revised the manuscript, and provided detailed explanations below. Please do not hesitate to contact us if there are any additional revisions we can make. We sincerely hope that this revised version is of higher quality and more appropriate for publication. All changes made in the paper have been highlighted in red font to aid further evaluation of our revised manuscript.

The whole manuscript has undergone thorough language editing (we made corresponding changes directly in the revised manuscript instead of marking them out in red as there are many small modifications).

Comments:

Line 48, 286: sentences need reconstruction.

56: "...is complicated." In this and the preceding paragraph, the present tense is appropriate. An exception would be a statement such as, "Smith et al. reported..."

Study participants: How many patients were approached for participation and how many declined? A flow chart of participant assignment in the study would be helpful. If available, report the characteristics of patients enrolled vs. not enrolled and evaluate for differences.

Data collection: The collection of anthropomorphic and blood pressure measurements should be described.

Response: Thanks for your comment. We apologize that we failed to upload the supplementary tables to the system. The sentence in Line 48 has been reconstructed to: “Surprisingly, a considerable variation was detected in GV, and whether other factors are involved is yet unknown, although the levels of the aforementioned GV-affecting clinical indicators of some T1D patients were comparable. ” (Line 50-53)

The sentence in Line 286 has been reconstructed to: “With the increasing use of CGM, several studies have revealed that GV may be involved in the emergence of diabetic complications [17–19], although the mechanism is not yet understood.” (Line 341-343)

Line 56 has been changed to “… is complicated.” (Line 60)

As you suggested, we have added a flow chart of the comprehensive analysis of metabolomics profiles of patients with different glycemic variability in the supplementary materials (Figure S1). And the characteristics of patients enrolled vs. not enrolled were compared in the supplementary materials (Table S4).

The collection of anthropomorphic and blood pressure measurements has been described in the text as follows: “Height and weight were measured using the electronic stadiometer and weighing scale. BMI was calculated as weight in kg / (height in m)2. Patients were asked to stand with their arms crossed on the contralateral shoulders for waist circumference measurement. The measuring tape should be snugly around the lateral aspect of each ilium at the mid-axillary line. The patients were asked not to smoke, drink strong tea or coffee within 30 min before blood pressure measurement, and empty their bladder. Blood pressure was measured using cuff pressurization after sitting in a chair with a backrest in a quiet environment for at least 5 min. ” (Line 98-105)

We show special gratitude for all your comments.

Round 2

Reviewer 1 Report

Thank you for the responses and revisions.

For point 6, if you have authentic standards, the library should be mentioned in the method section (e.g. the vendor and catalog number). And I am assuming the library is huge? since you have more than 5000 metabolites identified.

It is still hard for me to understand how plasma samples could have metabolites such as benzamide, or phthialic acid, which are all aromatic. Are their identifications confirmed from the authentic standards?

Thank you for the response on point 7, and I think you should include it in the manuscript, so readers are more clear.

Author Response

Response to Reviewer 1 Comments

Thank you very much for offering us a second opportunity to revise our manuscript. We really appreciate your insightful comments and meticulous revisions to our manuscript. We carefully considered your suggestions, revised the manuscript, and provided detailed explanations below. Please do not hesitate to contact us if there are any additional revisions we can make. All changes made in the paper have been highlighted in red.

Point 1: For point 6, if you have authentic standards, the library should be mentioned in the method section (e.g. the vendor and catalog number). And I am assuming the library is huge? since you have more than 5000 metabolites identified.

Response 1: Thanks for your comment. We apologize for our improper response during the first round of revision. After confirmation with the technician who conducted the LC-MS analysis of the serum samples in our study, we concluded that the metabolites were first identified by the exact molecular weight (primary annotation) and then obtained by matching MS/MS fragments with the HMDB, MassBank, LipidMaps, mzCloud and BioDeepDB (a self-built standard database of Suzhou PANOMIX Biomedical Tech Co., that conducted the analysis) (secondary annotation). Subsequently, a total of 5,597 metabolites were identified in the primary and secondary annotations (pos: 3,012, neg: 2,585), and 569 metabolites were identified in the secondary annotation (used for drawing the pie chart - Fig.1d). All subsequent analyses were based on the 569 metabolites identified by the secondary annotation. And we have corrected this in the text. (Line 176-186)

In addition, in the first response to point 6, our improper description regarding the metabolite identification "Finally, the metabolites were validated by comparing retention time and fragmentation pattern to authentic standards" may have led to a misunderstanding. We sincerely apologize for this. Actually, those 5,597 metabolites are the total metabolites identified in the primary and secondary annotations.

Point 2: It is still hard for me to understand how plasma samples could have metabolites such as benzamide, or phthalic acid, which are all aromatic. Are their identifications confirmed from the authentic standards?

Response 2: Thanks for your comment. We apologize for our inappropriate response in the first round of revision. After communication and discussion with the technicians who conducted this LC-MS analysis, we speculated that the discovery of exogenous metabolites or drugs may have contributed to the presence of some benzene and its substituted derivatives in serum samples. It is possible to identify non-human metabolites because the metabolites were identified through the matching and identification in several public databases mentioned above and this step is species-indistinguishable. However, our subsequent analysis was based on those 25 metabolites enriched in seven KEGG metabolic pathways with significant differences between the GV-H and GV-L groups. In the KEGG pathway analysis, we limited the species type to Homo sapiens (human). In addition to benzamide and phthalic acid, amino acids analogues and neurotransmitters such as N-Acetyl-D-phenylalanine and Phenylethylamine were also classified as benzene and its substituted derivatives.

If you feel that the current pie chart showing metabolites classification (Fig.1d) is really not suitable, we can delete it.

Point 3: Thank you for the response on point 7, and I think you should include it in the manuscript, so readers are more clear.

Response 3: Thanks for your suggestion. We have already included it in the manuscript as you suggested. (Line 258-265)

We show special gratitude for all your comments.

Reviewer 2 Report

Thank you for your revision of the manuscript. I have a number of remaining issues:

Methods: please add your description of the stepwise regression modeling to the methods section.

Discussion, line 356: I think the you mean that L-arginine is used as a substrate to synthesise polyamines. However, this is not what the line currently states, please rephrase.

Discussion, paragraph about spermidine (lines 369-380): I do not understand why increased oxidative stress would lead to higher levels of spermidine. Since this assumption underlies your statements about how spermidine levels are increased in people with higher GV, this is critically important to explain clearly. Furthermore, what is missing from this paragraph is whether or not this could be related to different intake of dietary L-arginine levels which could increase the substrate for spermidine levels.

Discussion, paragraph regarding L-methionine levels: given that the patient population here is people with T1D, who in general do not suffer from insulin resistance, why would stimulation of hepatic gluconeogenesis improve glucose variability. Furthermore, do you assume that there is residual insulin secretion capacity since a large part of the paragraph is about how L-methionine improves pancreatic beta cell function, which is of course nil or very low in people with T1D?

Discussion, paragraph about trehalose: I think it is critical to add that the levels of trehalose measured in the study were very low and that therefore the relationships with trehalose need to be treated with caution.

Discussion, final paragraph: You need to add that you have not measured dietary intake and that the difference in levels of the metabolites could at least partially be due to differences in dietary intake.

Lastly, I think it should be emphasised that it will be important to repeat these analyses in different cohorts: when comparing the results from both cohorts in table S3, it is clear that for spermidine and L-metionine, the results seem relatively consistent with regard to the size of the FC. However, for trehalose (which is the only other one that changes in the same direction in both cohorts), the degree of change is halved which indicates that the result may not be reliable. Currently, the discussion and conclusions (and the same sections in the abstract) do not reflect this uncertainty. This needs to be changed.

Author Response

Response to Reviewer 2 Comments

Thank you very much for offering us a second opportunity to revise our manuscript. We really appreciate your insightful comments and meticulous revisions to our manuscript. We carefully considered your suggestions, revised the manuscript, and provided detailed explanations below. Please do not hesitate to contact us if there are any additional revisions we can make. All changes made in the paper have been highlighted in red.

Point 1: Methods: please add your description of the stepwise regression modeling to the methods section.

Response 1: Thanks for your suggestion. We have already included it in the methods section as you suggested. (Line 207-212)

Point 2: Discussion, line 356: I think you mean that L-arginine is used as a substrate to synthesise polyamines. However, this is not what the line currently states, please rephrase.

Response 2: Thanks for your careful check. We apologize for our inappropriate description of the synthesis of polyamines. Thus, we have rephrased this sentence as: “Polyamines are a class of compounds that contain two or more amino groups and are primarily synthesized from L-arginine and ornithine, with arginine decarboxylase and ornithine decarboxylase serving as key enzymes. ” (Line 368-371)

Point 3: Discussion, paragraph about spermidine (lines 369-380): I do not understand why increased oxidative stress would lead to higher levels of spermidine. Since this assumption underlies your statements about how spermidine levels are increased in people with higher GV, this is critically important to explain clearly. Furthermore, what is missing from this paragraph is whether or not this could be related to different intake of dietary L-arginine levels which could increase the substrate for spermidine levels.

Response 3: We appreciate your insightful comment. To address your concern, we have reorganized the discussion of the current literature on spermidine levels and T1D, as well as our explanations of increased spermidine levels in patients with high GV: “Higher amounts of polyamines have been documented in both exocrine and endocrine pancreatic cells, which may lead to endoplasmic reticulum stress, oxidative stress, inflammatory response, and autophagy [23]. Specifically, polyamines have been shown to play a pathogenic role in T1D progression because they are known to play a critical role in the functioning of β-cells [24] and immune cells [25]. The inhibition of polyamine biosynthesis significantly delays diabetes incidence in NOD mice [26]. Moreover, polyamines have a time- and concentration-dependent inhibitory effect on the protein phosphatase activity of insulin-secreting cells in diabetes [27]. In vivo, spermidine inhibited HbA1c and lipid peroxidation [28]. However, a recent study on the role of spermidine treatment in T1D pathogenesis found that daily oral spermidine treatment in NOD mice resulted in a higher diabetes incidence, organ-specific spermidine accumulation, elevated peripheral inflammation, and a lower proportion of suppressive Tregs [29]. Reportedly, spermidine is linked to diabetic complications. Spermidine levels in the aqueous humor of patients with proliferative diabetic retinopathy (PDR) can be 3-4 times higher than in healthy controls [30]. Furthermore, in insulin-treated diabetic patients, an increase in intra-erythrocytic spermidine content has been linked to both diabetic nephropathy and severe retinopathy [31]. It has also been discovered that children with T1D have higher polyamine oxidase activity, which could lead to increased ROS production and subsequent oxidative damage [32]. Similarly, spermidine oxidase activity, another polyamine catabolic enzyme, is significantly lower in T1D patients compared to non-diabetics [33]. This highlights the fact that decreased activity of polyamine catabolic enzymes in T1D patients may promote polyamine accumulation.

 The mechanism of elevated spermidine levels in T1D individuals with high GV is yet to be elucidated. Given that oxidative stress brought on by GV would be more severe than hyperglycemia alone [34-38], one explanation is that spermidine excess is a reactive change brought on by reactive oxygen species (ROS). GV-induced oxidative stress is related to an increase in ROS generation, oxidative DNA damage, and a reduction in superoxide dismutase function [39]. Reportedly, higher ROS generated spermidine, which protected the cell from oxidative damage [40,41]. In this context, changes in spermidine concentration could be a consequence, and may not be the cause, of high GV. So far, however, there are only assumptions regarding the precise mode of action and the impact on GV of different spermidine concentrations. To support or refute any of the foregoing hypotheses, additional in vitro and in vivo mechanistic research are required. ” (Line 373-417)

Point 4: Discussion, paragraph regarding L-methionine levels: given that the patient population here is people with T1D, who in general do not suffer from insulin resistance, why would stimulation of hepatic gluconeogenesis improve glucose variability. Furthermore, do you assume that there is residual insulin secretion capacity since a large part of the paragraph is about how L-methionine improves pancreatic beta cell function, which is of course nil or very low in people with T1D?

Response 4: We appreciate your thoughtful comments to us. It is true as you said the study population consists of individuals with T1D who are insulin-dependent and only have poor endogenous insulin secretory capacity. To address your concern, we have added some discussion regarding the role of L-methionine in T1D pathogenesis, progression and glycemic control as follows: “The role of methionine in T1D pathogenesis and progression has also been indicated in several studies. In T1D rats, L-methionine also protected pancreatic b-cells by regulating FOXO1 expression [46]. According to the German BABYDIAB study, children who developed autoantibodies by the age of two had a 2-fold lower methionine content than those who developed autoantibodies later or autoantibody-negative controls [47]. Another Environmental Triggers for Type 1 Diabetes (MIDIA) study found that methionine levels in T1D progressors were lower with time, according to longitudinal changes in plasma metabolic profiles [48]. Methionine deficits were also detected in the metabolomics profile of the pre-T1D mice [49]. Furthermore, there is evidence that low methionine levels are associated with increased oxidative stress. Significant T1D-dependent increases in circulating oxidation products and decreases in methionine and cysteine levels were detected, indicating increased oxidative stress [50]. Methionine is a component of the transsulfuration process, which leads to glutathione production, an important intracellular antioxidant. Prior studies have shown that insulin-deprived T1D patients have lower rates of homo-cysteine-methionine remethylation and higher rates of transsulfuration compared to control subjects [51]. Accordingly, patients with poor glycemic control have lower glutathione pools and lower erythrocyte free cysteine level that can be synthesized from methionine [52]. Therefore, the decrease in circulating methionine may indicate an increased glutathione production. In the present study, patients in the GV-H group had significantly lower serum L-methionine levels than those in the GV-L group, and L-methionine was inversely correlated with HbA1c, FBG and GV parameters in correlation analysis. Consequently, we speculate that decreases in circulating methionine may serve as a potential biomarker for T1D patients with poor glycemic control and high GV. ” (Line 431-454)

Point 5: Discussion, paragraph about trehalose: I think it is critical to add that the levels of trehalose measured in the study were very low and that therefore the relationships with trehalose need to be treated with caution.

Response 5: Thanks for your comment. As you suggested, we have added some description about the uncertainty existing in the association between trehalose levels and GV in the discussion: “It can be seen that uncertainty exists regarding the association between elevated trehalose levels and high GV in T1D patients. The levels of trehalose reflected by the peak intensity in the present study were very low when compared to spermidine and L-methionine. Moreover, it is worth noting that in both the exploratory and validation sets, the FC values of spermidine and L-methionine were relatively consistent. The FC of trehalose in the validation set, however, was less than half that of the FC of trehalose in the exploration set, indicating that the difference of trehalose between the GV-H and GV-L groups may not be as reliable as that of spermidine and L-methionine. Therefore, additional in vitro and in vivo mechanistic studies are required to clarify the relationship between high trehalose levels and high GV. ” (Line 472-482)

Point 6: Discussion, final paragraph: You need to add that you have not measured dietary intake and that the difference in levels of the metabolites could at least partially be due to differences in dietary intake.

Response 6: Thanks for your comment. It is true as you said the dietary intake evaluation is important in final metabolites identification, so we have added this limitation to the last paragraph of the discussion: “...One other potential limitation of our study is that we were unable to assess the impact of dietary intake on the circulatory metabolome of T1D patients, which may have contributed to the differences in levels of the present identified metabolites. Future investigation into diet-metabolome interactions will be required to clarify the impact of dietary intake on glycemic control and GV levels of patients with T1D. ” (Line 489-494)

Point 7: Lastly, I think it should be emphasised that it will be important to repeat these analyses in different cohorts: when comparing the results from both cohorts in table S3, it is clear that for spermidine and L-methionine, the results seem relatively consistent with regard to the size of the FC. However, for trehalose (which is the only other one that changes in the same direction in both cohorts), the degree of change is halved which indicates that the result may not be reliable. Currently, the discussion and conclusions (and the same sections in the abstract) do not reflect this uncertainty. This needs to be changed.

Response 7: Thank you for your thoughtful consideration. To address your concern, we have emphasized the importance to repeat these analyses in T1D patients with different geographical and nutritional backgrounds in the abstract, discussion and conclusions.

Abstract: Line 26-31

Discussion: Line 472-482, 489-494

Conclusions: Line 502-516

We show special gratitude for all your comments.

Round 3

Reviewer 2 Report

Thank you for your revision, I have no further questions.